# Cell-Based Nanoparticles Delivery Systems for Targeted Cancer Therapy: Lessons from Anti-Angiogenesis Treatments

**DOI:** 10.3390/molecules25030715

**Published:** 2020-02-07

**Authors:** Paz de la Torre, María Jesús Pérez-Lorenzo, Álvaro Alcázar-Garrido, Ana I. Flores

**Affiliations:** Grupo de Medicina Regenerativa, Instituto de Investigación Sanitaria Hospital 12 de Octubre (imas 12), Avda. de Cordoba s/n, 28041 Madrid, Spain; torre-merino.paz@h12o.es (P.d.l.T.); mariapl.imas12@h12o.es (M.J.P.-L.);

**Keywords:** cancer, angiogenesis, hypoxia, nanoparticles, nanomedicine, nanotechnology, mesenchymal stem cells, exosomes, cell membrane coating

## Abstract

The main strategy of cancer treatment has focused on attacking the tumor cells. Some cancers initially responsive to chemotherapy become treatment-resistant. Another strategy is to block the formation of tumor vessels. However, tumors also become resistant to anti-angiogenic treatments, mostly due to other cells and factors present in the tumor microenvironment, and hypoxia in the central part of the tumor. The need for new cancer therapies is significant. The use of nanoparticle-based therapy will improve therapeutic efficacy and targeting, while reducing toxicity. However, due to inefficient accumulation in tumor sites, clearance by reticuloendothelial organs and toxicity, internalization or conjugation of drug-loaded nanoparticles (NPs) into mesenchymal stem cells (MSCs) can increase efficacy by actively delivering them into the tumor microenvironment. Nanoengineering MSCs with drug-loaded NPs can increase the drug payload delivered to tumor sites due to the migratory and homing abilities of MSCs. However, MSCs have some disadvantages, and exosomes and membranes from different cell types can be used to transport drug-loaded NPs actively to tumors. This review gives an overview of different cancer approaches, with a focus on hypoxia and the emergence of NPs as drug-delivery systems and MSCs as cellular vehicles for targeted delivery due to their tumor-homing potential.

## 1. Introduction.

In recent decades, the predominant strategy of cancer treatment focused on the tumor cell. However, chemotherapeutic agents have a broad toxicity profile and they do not greatly differentiate between cancerous and normal cells. Furthermore, as a consequence of continual treatment, the cancerous cell becomes resistant to drugs, leading to therapy failure.

Solid tumors can be assimilated to an organ that, in addition to proliferating tumor cells, includes stromal cells, infiltrating inflammatory cells, extracellular support matrix and blood vessels, which together constitute the tumor microenvironment [1]. Anti-angiogenic treatments represented a change in the strategy against cancer, since the target is no longer the tumor cell but the endothelial cell and, for the first time, the tumor microenvironment. The blockage of the formation of new vessels in tumors attempts to inhibit tumor growth and to prevent metastasis. Angiogenesis, the sprouting of new capillaries from pre-existing vessels, is an adaptive response of tumor cells which allows oxygen delivery to hypoxic regions in the tumor, thereby sustaining tumor growth [2]. However, the formation of tumor vasculature is a rapidly growing and highly disorganized process which results in high interstitial fluid pressure (IFP), hypoxia and low extracellular pH. These vascular abnormalities create a barrier to drug administration, and are the main cause of tumor multidrug resistance [3].

## 2. Anti-Angiogenesis Therapy: A Revealing History

Vascular endothelial growth factor (VEGF) is the pivotal molecule in angiogenesis and its expression in the primary tumor correlates with a greater risk of recurrence and poor prognosis in a variety of cancers [4]. Other molecules structurally related to VEGF, which bind to the same receptors have been identified, such as Placental Growth Factor (PLGF), VEGF-B, VEGF-C, VEGF-D and the viral homologue of VEGF, VEGF-E [5]. VEGF promotes the survival of endothelial cells, and their proliferation and migration.

The first antiangiogenic agent approved by the Food and Drugs Administration (FDA) and later by the European Medicines Agency (EMA) was bevacizumad (Avastin, Roche), a humanized monoclonal antibody anti-VEGF, which binds and neutralizes all VEGF isoforms. Bevacizumad therapy proved to be of less benefit than expected, causing side effects such as severe bleeding, hypertension and thromboembolic events. Combined with conventional chemotherapy, bevacizumab demonstrated a modest but significant increase in overall survival in patients with metastatic colorectal cancer [6].

Other factors and signaling pathways, which directly or indirectly influence the process of tumor angiogenesis, have also been targets of anti-angiogenic therapy. These include platelet-derived growth factor (PDGF), fibroblast growth factor (FGF), hepatocyte growth factor (HGF), integrins, cyclooxygenase (COX-2), metalloproteases MMP-2, MMP-9 and hypoxia-inducible factor (HIF-1). Angiogenic signaling has also been blocked by the inhibition of specific receptors such as VEGFR-1 and -2, c-Met and PDGFR-β, which are expressed in both tumor and endothelial cells [7]. Furthermore, several molecules that target more than one pathway have been designed. This is the case of Brivanib, a VEGF and FGF receptor tyrosine kinase inhibitor, approved for the treatment of colorectal and hepatocellular carcinomas [8]. Likewise, tyrosine kinase inhibitors (TKIs), by blocking the signaling of several growth factor receptors, hold a therapeutic advantage over monoclonal antibodies, as they can simultaneously block multiple angiogenic pathways and potentially have greater efficacy. Some examples are pazopanib (VEGFR, PDGFR, FGFR, and c-Kit inhibitor), sorafenib (VEGFR, PDGFR, Raf, c-Kit and Flt-3 inhibitor) and sunitinib (VEGFR, PDGFR and c-Kit inhibitor) which have approval for the treatment of patients with advanced cancer [9,10,11].

Despite promising results in preclinical studies, anti-angiogenic treatments have shown insufficient effectiveness in clinical use. Although anti-angiogenic drugs delayed the progression of the tumor, an improvement in the overall survival was not achieved, and tumors continued growing [12]. Either inherently or in acquired form, many tumors are resistant to anti-angiogenic treatments, making them less effective from a therapeutic perspective.

## 3. Limits of Anti-Angiogenic Therapy

### 3.1. Tumor Microenvironment Resistance

Resistance to anti-angiogenic drugs is a common problem found in the treatment of several carcinomas, such as breast, lung, ovarian, colorectal, kidney and liver, among others. The main cause of resistance is the redundancy in the angiogenic pathways. There are multiple examples both at the preclinical and clinical level that the inhibition of one or more proangiogenic factors results in the induction of others, which would lead to the restoration of angiogenesis [13]. As a response to anti-angiogenesis, tumor cells begin to produce alternative compensatory proangiogenic factors. Likewise, the tumor microenvironment has been implicated in tumor response to therapy and contributes to both intrinsic and acquired drug resistance [14]. Due to the anti-angiogenic treatment, there is an increase in intra-tumor hypoxia, which acts as chemoattractant for immune cells from bone marrow. Macrophages, neutrophils, and myeloid-derived suppressor cells have been shown to sustain angiogenesis by stimulating VEGF-independent pathways [15]. Moreover, these tumor-infiltrating myeloid cells are critical in establishing an immunosuppressive tumor microenvironment, which promotes tumor evasion and resistance to therapy [16].

### 3.2. Pro-Metastatic Effect of Anti-Angiogenic Therapies

Significantly, therapeutic inhibition of angiogenesis has been related to increased local invasiveness and distant metastasis [17]. Changes in tumor cells and in the tumor microenvironment due to the hypoxic condition generated by anti-angiogenic therapy seems to promote increased migration of tumor cells. The relationship of hypoxia with more aggressive metastatic cell behavior is well established [18]. Different tumor-dependent mechanisms, mainly driven by the hypoxia-inducible factor HIF-1, could be involved in metastasis, including the production of pro-metastatic proteins, the disruption of basement membranes by the secretion of proteolytic enzymes [19] or the alteration of the adhesion molecule pattern, promoting epithelial to mesenchymal transition [20]. Furthermore, increased hypoxia levels favor the recruitment of endothelial progenitor cells (EPCs) which promote the formation of a vascular, pre-metastatic niche [17].

## 4. New Objective: Targeting Hypoxia

Anti-angiogenesis therapy failure has revealed that overcoming tumor hypoxia must be a principal objective in the treatment of cancer. Tumor cells growing in a hypoxic environment quickly adapt and undergo genetic changes to resist hypoxia-induced cell death. Targeting hypoxia is fundamental not only to prevent metastasis, but also to overcome resistance toward conventional cancer therapies including chemotherapy, radiotherapy [21] and photodynamic therapy [22], as well as to improve the clinical efficacy of cancer immunotherapy [23]. Preventing hypoxic condition might become a suitable strategy, along with current anti-angiogenic therapies, to improve effectiveness and minimize unwanted effects. As described, HIF-1 is responsible for the tumor adaptive response to oxygen and nutrient depletion, promoting metabolic changes [24], acidosis, immunosuppression [25,26] and angiogenesis [27]. Different chemical compounds interfering at different molecular levels in HIF signaling have been used in both preclinical and clinical studies [28]. Several molecules have been approved by the FDA and the EMA for use in the treatment of cancer, although limitations in tumor accumulation and variable systemic toxicity have been reported [29,30]. Topotecan, a topoisomerase 1 inhibitor which indirectly inhibits HIF, is used in the treatment of metastatic ovarian carcinoma and as a second line treatment for small cell lung cancer [31]. At present, other HIF signaling inhibitors are being tested in various clinical trials, either as single agents or in combination with other agents for the treatment of advanced or refractory cancers [32].

## 5. The Emergence of Nanomedicine

In the last two decades, the use of drugs contained in nanoparticles (NPs) has emerged as a solution in cancer therapy to direct drug delivery to the tumor and to reduce systemic damage. NPs size range is between 1 to 100 nm and they allow the absorption of high quantities of drug due to a large surface area-to-volume ratio. Small molecules, peptides, proteins, DNA or interference RNA have been loaded into NPs to be delivered to tumors. For biomedical applications, NPs should have a series of characteristics such as biocompatibility, degradability, stability, delivery efficiency and sustained release. Different types of NPs, such as organic (liposomes and polymers), inorganic (metallic, metal oxide, ceramic, and quantum dots) and carbon-based NPs (fullerenes, nanotubes), have been used in the field of medicine (Figure 1), especially in the treatment and imaging of tumors [33]. Besides their ability to passively accumulate at the tumor site, nanomaterials can be readily functionalized in order to improve their active targeting and cellular internalization [34]. To increase the release of the payload at target sites at the right time, stimuli-responsive NPs have been designed [35]. Interestingly, there are nanoparticles designed to release conjugated cytotoxic drugs as a response to the special chemical conditions of the tumor microenvironment, such as acidosis (reviewed in [36]) or hypoxia [37]. Furthermore, nanoscale materials can be aimed to be delivered across traditional biological barriers in the body such as the blood–brain barrier or the dense stromal tissue of the pancreas [38,39]. The use of encapsulated forms of a drug can improve the pharmacodynamics and pharmacokinetic properties of the substance [40]. Nanoparticles have advantages over conventional anti-tumor drugs because they can be multi-functionally designed to pinpoint several targets in the tumor microenvironment. Nanomedicine appears as a very promising field in cancer therapy, as demonstrated by the large number of publications that refer to successful in vitro and preclinical proofs of concept.

Nanotechnology provides different strategies to relieve intra-tumor hypoxia (Table 1). Some approaches are directed towards tissue re-oxygenation, either through in situ oxygen supply or by promoting intra-tumor H_2_O_2_ decomposition. Another therapeutic approach is the administration of HIF-1 blockers. Nanomaterials have been designed to silence the gene expression of HIF-1 by antisense oligonucleotides or by interference RNA (RNAi). Likewise, the encapsulated form of several HIF-1 signal-interfering drugs present some benefits with respect to the free drug, minimizing toxicity and/or improving its pharmacokinetic behavior.

As with conventional drugs, the flow of nanomedicines into the tumor may be negatively influenced by the hypoxia of the tumor microenvironment despite the existence of the enhanced permeability and retention effect (EPR). EPR exists in solid tumors as a consequence of the abnormalities in their vasculature, which lead to a selective extravasation of nanometric molecules in tumors, where they may reach a much higher concentration than in normal tissues [52]. Nanomedicine has taken advantage of this unique phenomenon, but the extreme hypoxic condition in the central region of a large tumor mass can limit the EPR effect, and be a barrier for the entrance of NPs. Several methods have been reported to enhance EPR, such as hyperthermia to mediate vascular permeabilization in solid tumors [53,54], ultrasound-induced cavitation to modify tumor tissue [55,56], the application of nitric oxide (NO)-releasing agents to expand blood vessels [57] or the administration of anti-hypertensives to normalize blood flow [58]. Some of these methods have been implemented from the field of nanomedicine to minimize side effects. Thus, different types of responsive-nanoparticles were designed to produce tumor heating after photostimulation, magnetism, radiofrequency waves or ultrasound [59]. Regarding NO, nanotechnology devices could facilitate its therapeutic use, which is limited by its short half-life, instability during storage, and potential toxicity [60].

Lessons from the undesired consequences of blocking angiogenesis in tumor progression led to the hypothesis that normalization of the tumor vasculature is a better therapeutic option than its destruction [61]. Vessel normalization would transform the abnormal phenotype of tumor vessels into a phenotype which resembles the normal functional vessels by repairing the basement membrane and increasing the rate of pericyte coverage and, consequently, decreasing the vessel leakage. Correcting the abnormalities in tumor vessels would prevent a further increase in harmful intra-tumor hypoxia levels, allowing medication delivery which is dependent on efficient blood flow. Normalization of tumor vasculature is also seen as necessary to improve NPs’ delivery [62]. In clinical practice, low doses of anti-angiogenic treatments attempt to achieve a balance of anti- and pro-angiogenic factors to make tumor vessels into a more normal phenotype. Moreover, the normalization of the aberrant vasculature of the tumor can present added advantages. Treatment with low doses of anti-VEGFR2 antibody resulted in a less immunosuppressive tumor microenvironment by the polarization of tumor-associated macrophages, and recruitment and activation of CD8+ T lymphocytes in a murine breast cancer model [63]. The infiltration of CD8+ T cells in tumor was associated with a better prognosis in various types of cancer. Unfortunately, vascular normalization does not have an easy application in clinical settings because it is a transitory state, and is referred to as the “normalization window”. It lasts 1–2 days and, in this period, the cytotoxic effects of anti-tumor drugs are markedly increased. Therefore, it becomes critical to adjust the timing of the cytotoxic treatments to take advantage of this vascular normalization window.

From the field of nanomedicine, attempts have also been made to design particles to promote the normalization of tumor vasculature, such as gold nanoparticles used to provide human recombinant endostatin (rhES) in tumors by EPR to facilitate transient vessel normalization and improve antitumor therapeutic efficacy [64]. Another approach is the combination of an anti-angiogenesis treatment and chemotherapy in the same nanomedicine formulation. The cytotoxic drug paclitaxel (PTX) is loaded into lipid-derivative conjugates (LGCs) made of anti-angiogenic agents such as a low molecular weight heparin and gemcitabine to simultaneously restore the tumor vasculature and deliver the cytotoxic drug [65]. Despite promising results obtained at the preclinical level, there is no clinical experience with any of these nanoplatforms.

Decades of research have yielded only a few anticancer nanomedicines currently in clinical use [66]. Some formulations are hypoxia-limiting drugs such as DaunoXome or liposomal daunorubicine, a chemotherapeutic drug of the anthracycline family used in Kaposi sarcoma, and Onivyde or liposomal irinotecan, a topoisomerase I inhibitor used for the treatment of pancreatic cancer [67,68]. These nanomedicines improve the safety profile of the drugs, but, as with other nanopharmaceuticals, the efficacy shown in preclinical experiments is not achieved at clinical level.

## 6. The Limits of Nanomedicine in Clinical Applications

It is a fact that the clinical translation of nanomedicine for the treatment of cancer remains a great challenge. Regardless of the important contributions of nanotechnology to oncology in minimizing the toxic side effects of drugs, overall survival of patients has not improved. Several relevant questions must be addressed in order to improve the applicability of nanomedicine formulations to treat cancer, and this requires the understanding of the complexity and heterogeneity of human tumors and a deeper insight into nano–bio interactions.

For NPs to have clinical translation potential, there is a need to evaluate their safety and toxicity in humans and determine how large-scale manufacturing processes can introduce changes in this profile. Although the safety of many materials has been proven, as the complexity of nanoparticles increases by the use of synthetic compositions or by the addition of ligands or coatings, the in vivo behavior and the toxicological profile must be evaluated. The main safety concerns derive from direct cell toxicity, nanoparticles aggregation, long-term accumulation, hemolytic effects, and/or immunogenic behavior [69]. Toxicological evaluation of nanoparticles is based on an understanding of their in vivo distribution, metabolism and excretion [70].

To take advantage of nanomedicine, it is vital to optimize nanomaterial properties such as drug-loading capacity and/or capability of sustained release of the cargo in vivo, among others. Furthermore, it is essential to minimize the location of nanoparticles in healthy tissues and improve their delivery to the target organ. Increasing the efficiency in the delivery of nanoparticles to the tumor is considered the main goal in order to achieve real benefit [71].

The use of nanomedicine in cancer therapy has been supported by the existence of the EPR phenomenon; however, only a small percentage of systemically injected NPs accumulate in tumors (a median of 0.7% according to a wide meta-analysis study based on preclinical data) [72]. EPR seems to be an overestimated effect, as its understanding is based on the high EPR existing in fast-growing subcutaneous tumor xenografts in mice models. However, non-invasive imaging techniques applied to a small number of patients to determine the penetration and accumulation of nanoparticles in tumor, revealed that EPR is not a uniformly extended effect in solid tumors in humans [67]. Variability in vascular permeability, blood velocity, interstitial blood pressure, oncotic pressure, and complexity of the tumor stroma influence the movement of nanoparticles into and out of the tumor [73]. Additionally, physicochemical properties of nanoparticles, principally size and shape, also affect NPs extravasation and accumulation. In order to predict tumor susceptibility to EPR, and therefore, to benefit from the use of nanomedicine, some attempts have been made to characterize EPR-related genes, proteins or cell biomarkers (reviewed in [74]). Several studies have suggested the value of stratifying subpopulations of cancer patients according to their EPR relevance, in order to define the “right patients” to be treated by nanomedicine strategies in an analogous manner, as is being done in the development of other anti-cancer strategies [67].

As an alternative to passive accumulation, active targeting of nanoparticles is proposed in order to improve their tumor retention and to favor their uptake by the target cells. This strategy relies on the interaction between ligands conjugated onto the surface of nanoparticles (e.g., antibodies, peptides or carbohydrates) and their target. Target substrates can be surface receptors expressed by tumor cells or by other cells in the tumor microenvironment, secreted molecules, or even the physicochemical environment in the tumor. An additional advantage of actively targeted NPs could lie in their capacity to target disseminated locations throughout the body, such as metastatic lesions or hematological cancers where EPR does not exist [75]. However, several problems have made the use of ligand-targeted approaches at the clinical level, so far, negligible. These problems are the accessibility and expression of the target, the anatomical and physiological barriers to NPs delivery, as well as the lack of real knowledge about the toxicities of these complex formulations [70].

The targeting of nanoparticles to tumors, whether active or passive, must overcome physiological barriers to reach the tumor site once systemically administered. Whatever increases circulating lifetime by reducing clearance means an improvement in efficacy. The clearance of NPs by kidneys and their sequestration by reticuloendothelial organs are the main barriers affecting their bio-distribution, and therefore must be considered at the design stage. Renal elimination of nanoparticles is determined by their size, charge, shape and surface composition [76]. Recognition of NPs by immune cells and retention by the reticuloendothelial organs, such as liver, spleen or bone marrow, constitute the other major obstacles to the success of nanoparticle delivery, since they lead to premature elimination from the bloodstream. Having interacted with biological fluids, nanoparticles are exposed to active biomolecules, and diverse serum proteins non-specifically adhere onto their surface, forming a protein corona. There is an evident impact of protein corona in the fate and biological effects of nanoparticles [77,78].

Several surface-coating molecules such as polyethylene glycol (PEG) have been used to provide “stealth” properties to NPs during circulation [79]. Nonetheless, complement-related responses to PEG result in mild to severe hypersensitivity reactions in some susceptible individuals [80]. In addition, PEG-specific antibodies have been detected after the repeated administration of PEG-coated liposomes in the same animal [81]. These immunological responses may lead to altered pharmacokinetics and subsequent loss of efficacy of the treatment, and to potentially serious toxicities including anaphylaxis.

## 7. Biological Carriers to Deliver NPs

Anti-angiogenic treatments have revealed the relevance of the tumor microenvironment in the progression of the tumor. In the battle against cancer, it is mandatory to attack simultaneously on various fronts. Then, targeting tumor cells, normalizing tumor vasculature and overcoming hypoxia, among others, are vital to control the growth of solid tumors and to prevent metastasis. Nanomedicine appears as a valuable tool to achieve these goals, having the opportunity to design polyvalent NPs. Unfortunately, physiological barriers decrease NPs circulating lifetime and hinder their delivery to the tumor site. Additionally, the hypoxic region of the tumor constitutes an insuperable barrier resulting in an inefficient distribution of the NPs, and, as a consequence, a non-uniform release of drugs into the tumor. Furthermore, the potential toxicity of NPs, owing to their composition and/or to the nano-bio interactions, can compromise the feasibility of their use [82]. It is necessary to find solutions to overcome these problems without forgetting the great heterogeneity of human tumors. The encapsulation of nanoparticles into cell- or cell membrane-based systems can enable these issues to be addressed to some extent.

Mesenchymal stem cells (MSCs), exosomes and plasma membrane coating are biocompatible candidates to transport NPs to the tumor site, given their stability, non-cytotoxic effects, high drug carrying capacity, and low- or no-immunogenic profile. In addition to the load of anticancer drugs, these biological carriers can be engineered to express therapeutic peptides and proteins, and to transport RNAs or imaging agents, this enabling a more synergistic approach to anticancer therapy. As discussed below, mesenchymal stem cells and their exosomes have tumor tropism because tumor hypoxia is a potent mediator directing MSCs’ migration [83]. Although the use of these encapsulated formulas is in initial stages, they appear as a potential way to reach the impenetrable hypoxic core of solid tumors.

### 7.1. Mesenchymal Stem Cells as Carriers to Deliver NPs

The use of stem cells as cellular vehicles of drug-loaded nanoparticles seems to be a very promising strategy for targeting tumor tissues [84]. Different types of stem cells could be used as cellular vehicles, such as embryonic stem cells, adult stem cells or induced pluripotent stem cells. Within the adult stem cell group, the mesenchymal stem/stromal cells (MSCs) are the most common cell type used, given their several advantages (Table 2), such as their availability, easy isolation, non-immunogenicity and immunomodulatory properties [85]. MSCs have been isolated from many sources, such as bone marrow [86], adipose tissue [87], umbilical cord tissue [88], amniotic fluid [89] and placenta [90]. From a clinical point of view, MSCs are the first choice of stem cells for use in cancer therapies. It is well known that MSCs specifically migrate, home and survive in tumor sites without being incorporated into normal tissue [91,92]. MSCs can migrate towards a tumor, since these cells respond to tissue damage, hypoxia and inflammation, as found in tumor microenvironments (Table 2). This tropism property makes MSCs a promising strategy for drug delivery systems in cancer therapy [91,93,94]. MSCs home to tumor stroma because of their attraction to several growth factors, cytokines and proteases existing in the tumors [95]. Exploring this tropism of MSCs toward tumor sites to deliver gene therapy, drugs or nanoparticles to the tumors is a promising strategy in cancer therapy (Figure 2).

There are other advantages in using MSCs in cancer therapy, such as the fact that they can be genetically modified to serve as a vehicle for cancer gene therapy, with little or no impact on their biology [96]. There are many reports showing that MSCs engineered to express anti-proliferative [97], pro-apoptotic [98] or anti-angiogenesis [93] agents were successfully used for the treatment of several types of tumors (reviewed in [96]). In addition, naïve MSCs have anti-proliferative [91,99], pro-apoptotic [100] or anti-angiogenic [93] properties and have a low risk of malignant transformation after transplantation in vivo due to their limited proliferation capacity [90,101]. Although some studies have proven that MSCs have pro-tumorigenic effects, increasing tumor growth and metastasis [102], it is generally agreed that MSCs can be manipulated with anticancer genes to be used in cancer therapies (Figure 2). The use of viral and non-viral vectors for genetic modifications to MSCs has several drawbacks, such as transient gene expression and low transfection efficiency, along with a high risk of cell transformation [103].

MSCs can also incorporate small molecules of anti-tumor agents, such as paclitaxel or doxorubicin, and carry them to tumor sites (Figure 2). However, this strategy has some downsides such as the low loading capacity and the rapid diffusional clearance of the molecules out of cells. Additionally, anti-cancer drugs may have some cytotoxic effects on MSCs and result in their loss before arrival at tumor sites. The effects of chemotherapeutics on MSCs have been quite controversial, from a reduction in proliferation and apoptosis, to resistance while retaining proliferation and differentiation potential (reviewed in [104]). MSCs are resistant to the cytotoxic effects of paclitaxel via the inhibition of their proliferation, inhibition of apoptosis and induction of quiescence [105]. However, paclitaxel exposure does not up-regulate the expression of the trans-membrane pump P-glycoprotein 1 in MSCs, a mechanism by which cancer cells resist paclitaxel treatment [106]. Doxorrubicin at clinically used doses induces premature senescence of MSCs in vitro [107]. These senescent MSCs are functional but not proliferative, and are protected from doxorubicin-induced tumor transformation. The effect of doxorubicin on MSCs in vivo is contradictory, from resistance to reduced proliferation rates and apoptosis [104], and this may be due to the ex vivo culture conditions of MSCs and duration of treatments. Hence, long-term in vitro and in vivo studies are necessary to understand the mechanisms behind the influence of chemotherapy on MSCs.

The encapsulation of chemotherapy drugs into NPs increases the drug-loading capacity of MSCs while reducing potential toxic effects on MSCs (Figure 2). In case of toxicity, the incorporation of controlled release or stimuli-responsive nanoparticles may avoid the loss of MSCs during the process of migration and tumor homing, as well as ensuring that a therapeutic dose of the anti-cancer agent is released at the tumor site [108,109]. MSCs loaded with ultrasound-responsive mesoporous silica NPs selectively released the cargo when the stimulus was applied, both, in vivo and in vitro [35]. When loaded with doxorubicin, these NPs caused death to mammary cancer cells in vitro after ultrasound exposition. This stimuli-response strategy would significantly reduce undesired side effects of anticancer treatments. Although progress has been made in the design of NPs to introduce anti-cancer drugs to be transported by MSCs, this combined system MSC/NP is still in its initial stages. It has been reported that the type of NP and its size, as well as its concentration, incubation time, and the presence or absence of serum in the culture medium, could interact with and alter the physical and phenotypical properties of MSCs [33]. However, other studies suggested that MSCs loaded with NPs preserved their morphology, proliferation, migration and homing capacity [35,84,110]. Mesoporous silica nanoparticles did not inhibit the tumor-tropic capacity of MSCs and, when loaded with doxorubicin-NPs, showed in vitro and in vivo migration towards tumors and in vitro induction of cancer cell death. [110]. It is an important goal to achieve clinical efficacy while ensuring safety by using small amounts of nanomaterials. MSCs loaded with carbon nanotube-doxorubicin presented migratory capacity, active targeting and long-term apoptosis of lung cancer cells in vitro and in vivo with extremely low doses of the anti-cancer nano-drug, with no side effects [111]. The sustained release of paclitaxel encapsulated into PLGA nanoparticles does not affect the functional capacities of MSCs or their tumor tropism and increases survival of tumor-bearing rats while decreasing glioma tumor tissue [112].

The main advantage of using MSCs is their ability to infiltrate uniformly into tumor tissue and this ability will improve the intra-tumor distribution of anti-cancer drugs [113]. Paclitaxel-loaded poly(lactic-co-glycolic acid) (PLGA) NPs had little effect on MSC viability, did not affect their migration and differentiation potential, and presented a dose-dependent cytotoxicity against lung cancer cells in vitro and in vivo [114]. In vivo, the paclitaxel-NPs-MSCs platform accumulated in lung tumors and produced higher tumor growth inhibition and survival compared to injected paclitaxel encapsulated in NPs [115]. Another study using MSCs with paclitaxel-loaded PLGA NPs in a rat model of orthotopic glioma showed similar results [112]. Achieving a high payload capacity is critical to getting therapeutic drug concentrations to the tumor site. Covalent conjugation or the physical association of NPs to MSCs’ surface can significantly increase the cargo besides the drug-NPs loaded by endocytosis. Such nanoengineered MSCs demonstrated greater tumor inhibition and increased in vivo survival with reduced systemic toxicity when compared to free or NP-encapsulated drugs in a variety of tumors (reviewed in [103]). In addition to anti-cancer drugs, NPs could also serve as carriers of bioactive molecules such as DNA, mRNA or siRNA into MSCs (Figure 2). Gene transfection of MSCs with plasmids encoding cytosine deaminase and uracil phosphoribosyl transferase suicide genes was successfully attained using a polyethylenimine (PEI) coating of mesoporous silica nanoparticles [116]. These PEI-plasmid-NPs induced cell death to breast cancer cells without producing any significant toxicity to the vehicle MSCs [116]. This approach will provide a double therapeutic effect after transplantation, migration and homing to tumor sites of the PEI-NPs-engineered-MSCs; one effect will be from the plasmid transported outside the NP and another from the drug transported inside the NP [35]. This Trojan-horse strategy could significantly improve the efficacy of NP-MSC-based anti-cancer therapy.

Nanoparticle-based anti-angiogenesis systems have been developed and studied in preclinical models. The anti-angiogenic properties of several types of NPs such as gold nanoparticles, silica and silicate-based nanoparticles, diamond nanoparticles, nanoceria nanoparticles, silver nanoparticles and copper nanoparticles, have been reported [82,117]. Encapsulation of these NPs in MSCs would overcome some of the limitations and side effects, and selectively target the tumor site, enhancing the anti-angiogenic properties of the NPs. On the other hand, the use of anti-angiogenic factors onto biocompatible nanoparticles has recently attracted great interest. Paclitaxel inhibits tumor growth using both a direct inhibition of tumor cell proliferation and an inhibition of angiogenesis [118]. However, paclitaxel develops hematologic toxicities (i.e., leukopenia and neutropenia) and liver damage. MSCs loaded with paclitaxel-PLGA nanoparticles showed a selective accumulation into the lungs of tumor-bearing mice with respect to non-tumor mice. This system presented a maintained concentration of paclitaxel in plasma for longer, and developed a deposition of paclitaxel in the lungs with a significantly lower “off-target” deposition in liver and spleen. In addition, the effect of these nano-engineered MSCs was a reduced proliferation of tumor cells, reduced angiogenesis, and increased apoptosis in the tumor tissue, and a less severe leukopenia because of the very low dose of paclitaxel [115]. The encapsulation of bevacizumab onto mesoporous silica nanoparticles is another promising drug delivery system for improving anti-angiogenic therapy [119]. Since mesoporous silica nanoparticles are well tolerated by MSCs [110], the use of this this system will be an effective treatment in tumor therapies.

MSCs have high tropism for the hypoxic microenvironment of tumors [83], and it has been described that the hypoxic preconditioning of MSCs improves the migration and homing of MSCs [120]. Hypoxic preconditioning of MSCs loaded with PEG-superparamagnetic iron oxide NPs increased their migration toward gliomas and their trafficking across the blood–brain barrier. This study showed the role of hypoxia in the migration and homing abilities of MSCs and the use of an innovative systemic MSCs-based cell therapy for the treatment of aggressive tumors [121]. The migration of MSC is regulated by several cytokine/receptor pairs. Chemokine receptor-4 (CXCR-4) and its interaction with stromal cell-derived factor SDF-1 secreted on the surface of tumor cells is the most important set involved in MSCs tumor tropism [122]. The use of biodegradable polymeric nanoparticles to overexpress CXCR-4 in human adipose MSCs enhanced cell migration velocity and increased their co-localization within the hypoxic area of the tumor [123]. Human MSCs loaded with iron oxide-NPs showed an overexpression of epidermal growth factor receptor (EGFR) that resulted in an improved migration of the MSCs towards hypoxic area of the tumor [124]. In addition, iron oxide NPs improved the homing and anti-inflammatory abilities of MSCs without modifying their properties [125]. These results suggest that NP-engineered MSCs could serve as vehicles to deliver therapeutic agents into hypoxic areas of tumors to overcome drug-resistance.

Once the cells are transplanted in vivo, it is important to monitor the long-term fate of MSCs, their migratory capacity and their biodistribution, as well as their tumor penetration capacity. The conjugation of different types of nanoparticles to MSCs have successfully demonstrated feasibility of tracking the migration and intratumor localization of MSCs by non-invasive imaging techniques used in preclinical and clinical settings, such as magnetic resonance imaging (MRI), computed tomography (CT), ultrasound, optical imaging, positron emission tomography (PET) and single-photon emission computed tomography (SPECT). Adipose-derived MSCs loaded with mesoporous silica-coated manganese oxide NPs were efficiently monitored by MRI imaging over long periods after transplantation [126]. Adipose-derived MSCs can be also monitored by non-invasive CT imaging in vivo after labelling by PEG-coated gold NPs [127]. These gold NPs were visible at the transplantation site for as long as four weeks with no loss in signal. The authors were able to quantify the number of visualized cells as a function of the CT value obtained. These results are very important to quantify the migratory and homing abilities of MSCs into tumor sites. Migratory capacity and tumor tropism toward malignant glioblastoma of both bone marrow and placenta-derived MSCs was demonstrated by in vivo MRI tracking after labeling with superparamagnetic iron oxide (PEG-SPIO)-NPs [121,128]. Although several nanoparticles have been designed for diagnostic and in vivo imaging, the optimal type of formulation for cell tracking in vivo does not, as yet, exist [129]. Further studies are necessary to use nanoparticles for diagnostic and imaging purposes in the field of oncology. The main challenges to be overcome are biocompatibility and an improvement in the synthesis process. This approach would allow the in vivo tracking and biodistribution of MSCs as carriers of therapeutic agents and would provide information about tumor-targeted accumulation, drug release and long-term drug efficacy. Such information could contribute to a model of personalized medicine and patient individualization.

Although there are several clinical trials with MSCs to treat several pathologies, their use in humans may cause some concerns owing to their potential to promote tumor growth, angiogenesis and fibrinogenesis [93]. There are other studies showing the anti-tumor and anti-angiogenic properties of MSCs [103], but whether MSCs facilitate or inhibit tumor growth remains controversial (Table 2). Therefore, a deeper understanding of the molecular and cellular interactions between MSCs and the tumor microenvironment is required.

### 7.2. Exosomes as Carriers to Deliver NPs

Exosomes have recently emerged as possible natural carriers of therapeutic agents for cancer therapy. These are small extracellular vesicles (30–150 nm in diameter) released from cells after the fusion of an intermediate endocytic compartment, the multivesicular body (MVB), with the plasma membrane. Exosomes are secreted by most eukaryotic cells and have been recognized as important messengers in cell-to-cell communication through the transfer of macromolecules such as lipids, proteins, and nucleic acids (mRNAs, tRNAs, long, noncoding RNA, microRNAs and mitochondrial DNA). Exosomes have been implicated in physiological processes but also in diseases such as neurodegenerative diseases [130], heart failure [131], liver disease [132], or cancer [133]. The proteomic and RNA content of exosomes from different cellular sources has been analyzed (www.exocarta.org) [134]. In addition to several conserved proteins that are common to most exosomes regardless of their origin, such as tetraspanins (CD9, CD63 and CD81) and heat shock proteins (HSP60, HSP70 and HSP90), among others, exosomes express tissue-specific proteins which reflect their originating parental cell [135]. Recently, it has been shown that exosomes have intrinsic homing capabilities similar to their original cells [136] and recapitulate their biological activity so that, in some cases, exosomes may be an alternative to cell therapy, avoiding the adverse effects of intravenous administration of cells.

The therapeutic properties of exosomes in clinical use can rely on three components: vesicle, load and/or surface decors. Exosomes are “natural nanoparticles” delivering an array of proteins and nucleic acids, and they can also be engineered to improve tumor recognition and killing properties in order to increase the effectiveness of cancer therapy [137]. The efficient encapsulation of different types of nanoparticles into exosomes for therapeutic and diagnostic purposes in cancer is also possible [138,139,140]. Compared to synthetic nanomaterials, a first advantage of the use of exosomes is their innate biocompatibility so they may be less immunogenic or cytotoxic (Table 2). Exosomes are large enough to avoid being cleared rapidly by kidneys and, in some cases, small enough to escape the capture of the mononuclear phagocyte system and to take advantage of the EPR effect accumulating in tumors. Exosomes are formed by a lipid bilayer delimiting an aqueous core, and this allows the upload of drugs of both, hydrophobic and hydrophilic natures, thus increasing their versatility [141]. Some concerns in the scalable production of exosomes for clinical use are the establishment of optimal culture conditions, product purity, batch uniformity and storage conditions (Table 2), among others, to ensure that the exosome-based product meets the expected quality and efficiency [142]. The relevant issues are the optimal dose, the timing of administration, and the route of injection to achieve maximal efficacy and minimize adverse effects [143]. The fate of exosomes can be monitored in vivo by labeling them with NPs that are detected by non-invasive imaging techniques such as magnetic resonance imaging (MRI), computed tomography (CT) or magnetic particle imaging (MPI) [144,145]. These techniques have advantages and disadvantages according to their sensitivity, specificity, penetration, radiation and spatial resolution. Further developments need to be investigated to obtain more efficient, biocompatible and quantifiable exosome labeling and imaging techniques with the aim of translating exosome therapy to the clinic [144,146].

Although many types of cells in the body produce exosomes, MSCs are one of the most prolific (Figure 2), and, therefore, are more suitable for the mass production of exosomes for drug delivery [147]. Clinical trials of MSC-derived exosomes that are currently in progress focus on gene delivery, regenerative medicine, and immunomodulation [136]. As expected, MSC-derived exosomes have intrinsic homing capabilities similar to those of MSCs and, in the treatment of cancer, can penetrate the tumor site [148]. In a similar way, hypoxia could also be a target for MSC-derived exosomes. Very interestingly, in hypoxia studies it has been published that hypoxic cancer cells avidly uptake exosomes, which have been produced in hypoxic conditions [149]. Culturing MSCs in hypoxic conditions would not only produce hypoxia-conditioned exosomes but also lead to an increase in exosome production, as described [150]. The effects of native MSCs-derived exosomes in cancer remain controversial and further analysis is required. Observed pro-tumor or anti-tumor effects are supposed to rely on cell culture conditions, on the methods to promote vesicle formation and on the tumor model used [151]. Although a cancer suppression ability by MSCs native exosomes from adipose tissue has been reported in vitro [152] and in vivo [153], genetic modification is the commonly used strategy in MSCs-derived exosome-based cancer therapy through transfection of diverse miRNAs or siRNAs. These genetically modified exosomes have provided the reduced viability of cancer cell lines, growth inhibition of tumor xenografts and/or prolonged survival in mice cancer models (reviewed in [137]). On the other hand, active drugs can be incorporated into exosomes from primed MSCs, resulting in in vitro antitumor effects [154].

### 7.3. Cell Membrane-Coated Nanoparticles

Another strategy to deliver nanomedicines into tumors is the use of cell membrane-coated NPs [155]. These systems use the plasma membrane of different types of cells, such as red blood cells [156], leukocytes [157], macrophages [158], platelets [159], stem cells [160], bacterial [161] or cancer cells, [162] to coat the NPs. Depending on its origin, the membrane could provide different in vivo biological behavior to these cell membrane-based NPs. For example, blood and immune cell membranes could be responsible for an extended systemic circulation and for avoiding immune clearance, while the cancer cell membrane will provide tumor targeting (Table 2). In addition, using the fusion of cell membranes from different sources, the hybrid cell membrane-based NPs obtain multiple functionalities [163].

The nanoparticle used in the core would be designed according to its future application, such as anti-cancer drug delivery, tissue imaging or photothermal therapy [163,164]. It has been proven that these membrane-coated NPs accumulate preferentially at tumor sites, improving their efficacy while reducing their toxicity [163]. A macrophage-biomimetic drug delivery system with anti-angiogenesis properties was developed by coating PLGA-NPs with macrophage membrane [165]. PLGA-NPs were loaded with saikosaponin D, a compound which exhibits potential anticancer therapeutic properties. The authors showed that these cell-membrane engineered NPs effectively inhibited tumor growth and metastasis of breast cancer in vitro and in vivo through the inhibition of angiogenesis. Polymeric NPs loaded with the anticancer drug paclitaxel were coated with red blood cell (RBC) membranes. This RBC-biomimetic drug delivery system significantly inhibited tumor growth and suppressed lung metastasis [166]. Although the angiogenesis inhibition by paclitaxel was not evaluated, these RBC-mimetic NPs seem to be an efficient system for cancer therapy.

To overcome tumor hypoxia and improve the therapeutic effects of anti-cancer treatments, several attempts based in membrane-camouflage have been made. Platelet membranes as nanocarriers were co-loaded with tungsten oxide (W_18_O_49_) nanoparticles and metformin (PM-W_18_O_49_-Met NPs) to treat lymphoma tumors. In this system, metformin reduced tumor oxygen consumption to alleviate tumor hypoxia, enhancing the therapeutic effects of W_18_O_49_ mediated by reactive oxygen species (ROS) and heat generation [167]. PM-W_18_O_49_-Met NPs significantly inhibited tumor growth and induced apoptosis in lymphoma tumors in vitro and in vivo. Platelet membranes provide immune evasion and active adhesion to tumor cells mediated by the interaction of platelet P-selectin with ligands expressed on tumor cells [168]. Other approaches used RBCs as the source of biomimetic membranes. RBCs are one of the most abundant cell types in the body and, as indicated above, their use as biomimetics provides immune escape and a long blood circulation time for NPs. In addition, enzymatically active catalase in the RBC membrane [169] could metabolize tumor endogenous H_2_O_2_ and ameliorate tumor hypoxia. PLGA-NPs coated with RBC membranes were engineered to co-deliver the chemotherapeutic agent curcumin, and the hypoxia-activated molecule, tirapazamine [170]. These drug-loaded and coated NPs induced apoptosis via the generation of reactive oxygen species and consequent DNA damage, suggesting the potential of the present system to circumventing hypoxic solid tumors. In another approach, the membrane from red blood cells was used to encapsulate nanoparticles consisting of perfluorocarbon inside PLGA [171]. This nanomimetic approach could provide an efficient supply of oxygen to the tumor site. Recently, encapsulated Ag_2_S quantum dots in RBC membranes have been used as a sonosensitizer to generate ROS under ultrasonic stimulation [172]. This design takes advantage of ultrasound to promote tumor blood flow, improving hypoxic conditions and enhancing the sonotherapic effect of the system. In combination with oral anti-tumor drugs, this approach significantly increased survival of tumor-bearing mice.

Although encouraging results have been found using cell membrane-coated nanoparticles to treat tumor angiogenesis and hypoxia, this field needs further lines of investigation such as improvement of the production process to increase the yield and decrease the batch-to-batch variability (Table 2), as well as a better knowledge of the proteins present in cell membranes to avoid unexpected adverse immune reactions.

## 8. Conclusions

In cancer therapy, new targets are needed, and the microenvironment provides a wide range of elements to be targeted. Anti-angiogenic therapies proved lesser benefit than expected and revealed the relevance of overcoming intra-tumor hypoxia. Hypoxia is a characteristic abnormality of tumor responsible for the resistance towards conventional cancer therapies. Correction of hypoxia levels and abnormalities in tumor vessels may improve medication administration and the outcome of treatments. Taking into account the results reviewed in this article, the most effective strategy in cancer therapy appears to be the simultaneous targeting of several processes, such as those involved in tumor cell proliferation, tumor angiogenesis and tumor hypoxia. Therefore, to obtain real clinical benefits it is necessary to design combined treatments.

Nanotechnology emerges as a fundamental tool in the design of multifunctional anticancer agents. Nanoparticles will offer an improvement with regard to drug cargo, and targeting and reducing toxicity, as compared to traditional chemotherapeutic agents. However, NPs present several problems such as poor penetration inside the tumor and rapid clearance by the reticuloendothelial system.

The use of MSCs as vectors for the delivery of anticancer agents is a very promising strategy due to their tropism for the tumor microenvironment. Moreover, they are easily available, non-immunogenic and can be manipulated in vitro. To date, there are several studies suggesting that unmodified MSCs can exhibit both anti- and pro-tumor properties. Further research is needed to understand the biological differences between MSCs obtained from different sources and to standardize the culture conditions in order to improve the safe use of this approach. Two different strategies can be used to combine MSCs and NPs. Firstly, MSCs are engineered by loading NPs with anti-cancer drugs to be released into tumor sites. Secondly, drug-loaded NPs are genetically engineered by a gene vector to produce antitumor proteins and later introduced into MSCs. This system acts as a “Trojan Horse” to deliver two different therapeutic agents to targeted sites. Besides the challenges of biocompatibility and improvement in the synthesis process of NPs, further studies are necessary to unravel the role of MSCs in facilitating or inhibiting tumor growth. Therefore, the use of the MSC/NP system needs additional study before it can be used clinically in humans.

MSC-derived exosomes can be an alternative to the use of MSCs. Exosomes are endogenous nanoparticles delivering biomolecules and can also be engineered to be used in cancer therapy. However, the anti- or pro-tumor properties of native MSCs have also been observed in isolated exosomes. More studies are needed to evaluate MSCs exosomes’ migration and homing abilities. Overall, future studies should be carried out to understand exosomes’ and NPs’ interaction, in vivo biodistribution and interrelation with the tumor microenvironment.

The newest strategy is to use membranes from different types of cells to protect the NPs, but this approach needs further development.

Although nanomedicine-related technologies need additional improvements, it is reasonable to assume that the best approach to treat cancer in the future is to combine anti-tumor, anti-angiogenic and anti-hypoxic agents. These treatments could be administered simultaneously by targeting delivery methods such as NPs, NP-loaded MSCs, exosome-natural NPs, NP-loaded exosomes and NP-loaded membranes for a more selective and effective method of treatment.

## Figures and Tables

**Figure 1 molecules-25-00715-f001:**
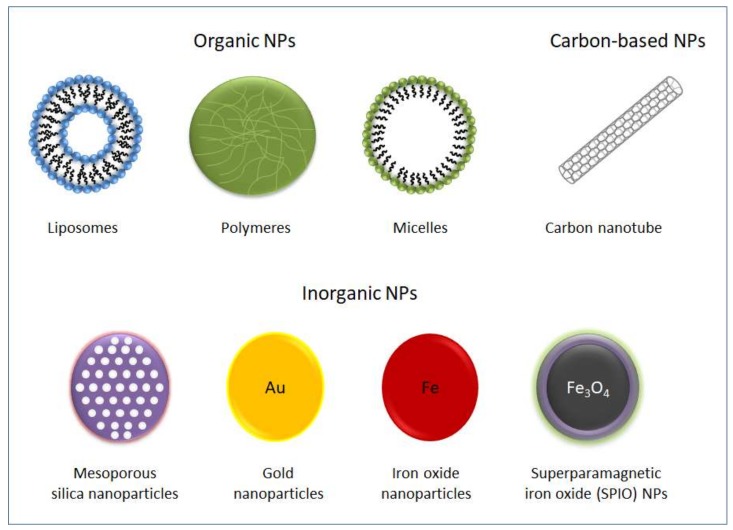
Types of nanoparticles commonly used for biomedical applications.

**Figure 2 molecules-25-00715-f002:**
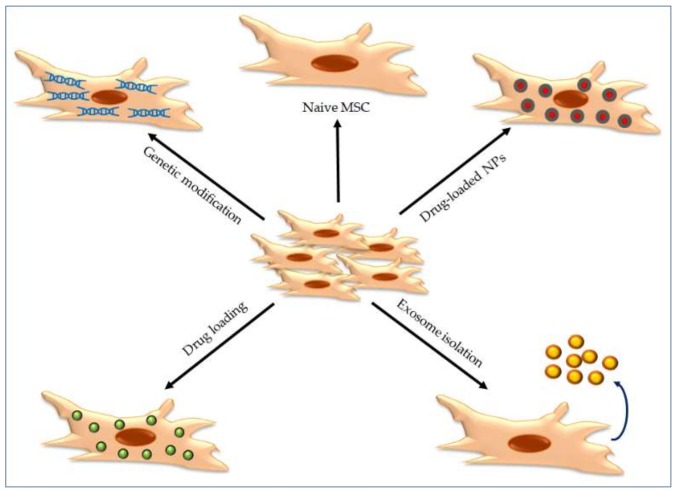
Mesenchymal stem cell (MSC)-based strategies for targeted-cancer therapy. MSCs can be used an anti-cancer agents due to their tumor-tropic properties, and their anti-proliferative, pro-apoptotic or anti-angiogenic properties (naïve MSCs). MSCs can be genetically modified to express suicide or anti-tumor genes. MSCs can incorporate small molecules of anti-tumor agents and they have been used as cellular vehicles of NPs. In addition, MSC-derived exosomes can be used as drug-delivery tools.

**Table 1 molecules-25-00715-t001:** Nanotechnology strategies against hypoxia.

Categories	Cargo	Type of Nanoparticle	Mode of Action	Ref.
**O_2_ carriers**				
Perfluorocarbonand derivatives	Perfluorocarbon Perfluorooctane Perfluorohexane Perfluorocarbon	PLGA-PEG emulsion Hollow microparticles Liposomes Hollow Bi_2_Se_3_ NPs	Rapid release of O_2_ by hydrolysis	[41] [42] [43] [44]
Ultrasound-based carrier	Oxygen	Microbubbles	Ultrasound controlled release and imaging by ultrasonography	[45]
**H_2_O_2_ catalysis platforms**				
	MnO_2_ Catalase	UPCNPs Liposomes	Decomposition of H_2_O_2_ into O_2_ and H_2_O	[46] [47]
**Inhibitors of HIF-1 signaling**				
	Camptothecin Topotecan HIF-1 siRNA HIF-1 ASO	Cyclodextrin-based polymer Liposomes PEGylated ε-polylysine copolymer Liposomes	Topoisomerase I inhibition Topoisomerase I inhibition Reduction of HIF-1 levels Reduction of HIF-1 levels	[48] [49] [50] [51]

Abbreviations: PLGA, Poly(lactic-co-glycolic acid); PEG, Polyethylene glycol; US, ultrasound; UPCNPs, up-conversion nanoparticles; ASO, antisense oligonucleotide.

**Table 2 molecules-25-00715-t002:** Advantages and disadvantages of the different cell-based delivery systems.

Cell-Based Strategy	Advantages	Disadvantages
**MSCs**	Easy isolation from accessible sources Easy culture in vitro No immunogenicity Tissue regeneration capacity Tumor tropism Migration ability into the site of damage Homing capacity	Uncertain tumorigenic effect High retention in lungs after systemic administration Possible occlusion of microvessels after systemic administration Repeated injection may result in production of alloantibodies
**EXOSOMES**	High stability in physiological and pathological conditions Unlikely to be immunogenic Small and relatively homogeneous size Intracellular delivery of cargo by fusion of membranes Able to cross natural barriers such as blood–brain barrier Autologous use allowing personalized medicine	Need of standardized protocols for isolation and purification Need of adequate characterization of cell of origin Undesired effects due to exosome components themselves Lack of standardized mass production protocols
**PLASMA MEMBRANE-COATING**	Provide biocompatibility to nanoparticles Immune escape and longer circulation lifetime High versatility Easy functionalization	Need of techniques for large-scale cell culture Need of high-yield methods for membrane derivation Lack of knowledge about all membrane components

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
