# Peer review of "Cell-Based Nanoparticles Delivery Systems for Targeted Cancer Therapy: Lessons from Anti-Angiogenesis Treatments"

_molecules, 2020, doi:10.3390/molecules25030715_

Round 1

Reviewer 1 Report

The review focuses on the anti-angiogenesis treatment, showing the conventional practices and related problems and then moves to the description of innovative cell or extracellular-vesicles based nanoparticles. However, despite the focus of the Review is clear, the description of the single paragraphs is not all consistent and focused on the specific topic.

The use of all the natural carriers, i.e. MSC, exosomes, and Cell membrane-coated nanoparticles should be more focused on the hypoxia and anti-angiogenic treatment.

Paragraph 7 "Use of Nanoparticles for in vivo imaging of MSCs after use as anti-tumor cellular vehicle" is very short, not well positioned in the text logic course and thus has to be broadened and better integrated with the logic structure of the review.

Furthermore, more schemes, figures/data taken from the cited papers as well as three resuming tables concerning each sub topic (MSC, exosomes, and Cell membrane-coated nanoparticles) and comparing in each cited references sub-case in terms of their use and the role of each component have to be inserted.

Other comments:

page 5, line 178-179: the authors should also mention the role played by ultrasound in helping drug or nanoparticles extravasation to reach the tumor site. In particular, the work of prof. Constantin Coussios should be mentioned. Here some examples:

Ultrasound Med Biol. 2011 Nov;37(11):1838-52. doi: 10.1016/j.ultrasmedbio.2011.08.004. Epub 2011 Oct 2. "Cavitation-enhanced extravasation for drug delivery".   Ultrasound Med Biol. 2019 Apr;45(4):954-967. doi: 10.1016/j.ultrasmedbio.2018.10.033. Epub 2019 Jan 14. "Microbubbles, Nanodroplets and Gas-Stabilizing Solid Particles for Ultrasound-Mediated Extravasation of Unencapsulated Drugs: An Exposure Parameter Optimization Study".

lines 279-285: it is fundamental to elucidate in the review the mechanisms of why MSC are not susceptible to the therapeutic content carried by the internalized NP while the cancer target cell are and if there are proofs and drawbacks of this method.

line 341: The authour mention just one example of encapsulation of Gold nanoparticles into exosomes for therapeutic and diagnostic purposes against cancer. Actually recently new papers were published about this topic using Metal Organic Framework (MOF) or Zinco oxide nanoparticels, which can be cited:

Chem. Mater. 2017, 29, 19, 8042-8046 "Exosome-Coated Metal–Organic Framework Nanoparticles: An Efficient Drug Delivery Platform"

Nanomedicine 2019 14:21, 2815-2833 "ZnO nanocrystals shuttled by extracellular vesicles as effective Trojan nano-horses against cancer cells"

Minor comments on English:

line 181: change "to conduce" in "to conduct " line 379: change "such us2 in "such as"

Author Response

Response to Reviewer 1 Comments

Point 1. The use of all the natural carriers, i.e. MSC, exosomes, and Cell membrane-coated nanoparticles should be more focused on the hypoxia and anti-angiogenic treatment.

Response 1. We gratefully acknowledge the comments made by the reviewer, aimed to improve the quality of the work.

According to the referee´s comment, we have included some paragraphs in each subsections to focus the use of MSC, exosomes and cell membrane-coated nanoparticles in hypoxia and angiogenesis treatments. We have also extended some paragraphs to highlight the description of hypoxia and angiogenesis.

Pages 7-8, lines 284-296: Anti-angiogenic treatments have revealed the relevance of tumor microenvironment in the progression of the tumor. In the battle against cancer, it is mandatory to attack simultaneously on various fronts. Then, targeting tumor cells, normalizing tumor vasculature and overcoming hypoxia, among others, are vital to control the growth of solid tumors and to prevent metastasis. Nanomedicine appears as a valuable tool to achieve these goals having the opportunity to design polyvalent NPs. Unfortunately, physiological barriers decrease NPs circulating lifetime and hinder their delivery to the tumor site. Additionally, the hypoxic region of the tumor constitutes an insuperable barrier resulting in an inefficient distribution of the NPs, and as a consequence, a non-uniform release of drugs into the tumor. Furthermore, potential toxicity of NPs, owing to their composition and/or to the nano-bio interactions, can compromise feasibility of their use [82]. It is necessary to find solutions to overcome these problems without forgetting the great heterogeneity of human tumors. Encapsulation of nanoparticles into cell- or cell membrane- based systems can enable the address of these issues to some extent.

Page 8, lines 302-305: As discussed below, mesenchymal stem cells and their exosomes have tumor tropism because tumor hypoxia is a potent mediator directing MSCs migration [83]. Although the use of these encapsulated formulas is in their initial stages, they appear as a potential way to reach the impenetrable hypoxic core of solid tumors.

Page 11, lines 409-433: Nanoparticle-based anti-angiogenesis systems have been developed and studied in preclinical models. Anti-angiogenic properties of several types of NPs such as, gold nanoparticles, silica and silicate-based nanoparticles, diamond nanoparticles, nanoceria nanoparticles, silver nanoparticles and copper nanoparticles have been reported [82, 117]. Encapsulation of these NPs in MSCs would overcome some of the limitations and side effects, and selectively target the tumor site enhancing the anti-angiogenic properties of the NPs. On the other hand, the use of anti-angiogenic factors onto biocompatible nanoparticles has recently attracted great interest. Paclitaxel inhibits tumor growth using both, a direct inhibition of tumor cell proliferation and an inhibition of angiogenesis [118]. However, paclitaxel develops hematologic toxicities (i.e. leukopenia and neutropenia) and liver damage. MSCs loaded with paclitaxel-PLGA nanoparticles showed a selective accumulation into lungs of tumor-bearing mice with respect to non-tumor mice. This system presented a maintained concentration of paclitaxel in plasma for longer, and developed a deposition of paclitaxel in the lungs with a significantly lower “off-target” deposition in liver and spleen. In addition, the effect of these nano-engineered MSCs was a reduced proliferation of tumor cells, reduced angiogenesis, and increased apoptosis in the tumor tissue, and a less severe leukopenia because of the very low dose of paclitaxel [115]. Encapsulation of bevacizumab onto mesoporous silica nanoparticles is another promising drug delivery system for improving anti-angiogenic therapy [119]. Since mesoporous silica nanoparticles are well tolerated by MSCs [110], the use of this this system will be an effective treatment in tumor therapies.

MSCs have high tropism for the hypoxic microenvironment of tumors [83], and it has been described that hypoxic preconditioning of MSCs improves migration and homing of MSCs [120]. Hypoxic preconditioning of MSCs loaded with PEG-superparamagnetic iron oxide NPs increased their migration toward gliomas and their trafficking across blood-brain barrier. This study showed the role of hypoxia on the migration and homing abilities of MSCs and the use of an innovative systemic MSCs-based cell therapy for treatment of aggressive tumors [121].

Page 11, lines 433-443 (this paragraph was already included in the original version): Migration of MSC is regulated by several cytokine/receptor pairs. Chemokine receptor-4 (CXCR-4) and its interaction with stromal cell-derived factor SDF-1 secreted on the surface of tumor cells is the most important set involved in MSCs tumor tropism [122]. The use of biodegradable polymeric nanoparticles to overexpress CXCR-4 in human adipose MSCs enhanced cell migration velocity and increased their co-localization within the hypoxic area of the tumor [123]. Human MSCs loaded with iron oxide-NPs showed an overexpression of epidermal growth factor receptor (EGFR) that resulted in an improved migration of the MSCs towards hypoxic area of the tumor [124]. In addition, iron oxide NPs improved homing and anti-inflammatory abilities of MSCs without modifying their properties [125]. These results suggest that NP-engineered MSCs could serve as vehicles to deliver therapeutic agents into hypoxic areas of tumors to overcome drug-resistance.

Page 13, lines 515-521: As expected MSC-derived exosomes have intrinsic homing capabilities similar to those of MSCs and in the treatment of cancer can penetrate the tumor site [148]. In a similar way, hypoxia could also be a target for MSC-derived exosomes. Very interestingly, in hypoxia studies it has been published that hypoxic cancer cells avidly uptake exosomes, which have been produced in hypoxic conditions [149]. Culturing MSCs in hypoxic conditions would not only produce hypoxia-conditioned exosomes but also lead to an increase in exosome production, as described [150].

Pages 13-14, lines 544-576: A macrophage-biomimetic drug delivery system with anti-angiogenesis properties was developed by coating poly(lactic-co-glycolic acid) nanoparticles (PLGA-NPs) with macrophage membrane [165]. PLGA-NPs were loaded with saikosaponin D, a compound which exhibits potential therapeutic properties for cancer therapy. The authors showed that these cell-membrane engineered NPs effectively inhibited tumor growth and metastasis of breast cancer in vitro and in vivo through inhibition of angiogenesis. Polymeric NPs loaded with the anticancer drug paclitaxel were coated with red blood cell (RBC)-membranes. This RBC-biomimetic drug delivery system significantly inhibited tumor growth and suppressed lung metastasis [166]. Although the angiogenesis inhibition by paclitaxel was not evaluated, these RBC-mimetic NPs seem to be an efficient system for cancer therapy.

To overcome tumor hypoxia and improve the therapeutic effects of anti-cancer treatments, several attempts based in membrane-camouflage have been made. Platelet membranes as nanocarriers were co-loaded with tungsten oxide (W18O49) nanoparticles and metformin (PM-W18O49-Met NPs) to treat lymphoma tumors. In this system, metformin reduced tumor oxygen consumption to alleviate tumor hypoxia enhancing the therapeutic effects of W18O49-mediated by reactive oxygen species (ROS) and heat generation [167]. PM-W18O49-Met NPs significantly inhibited tumor growth and induced apoptosis in lymphoma tumors in vitro and in vivo. Platelet membrane provide immune evasion and active adhesion to tumor cells mediated by interaction of platelet P-selectin with ligands expressed on tumor cells [168]. Other approaches used RBC as the source of biomimetic membranes. RBC are one of the most abundant cell types in the body and, as above indicated, their use as biomimetics provide immune escape and long blood circulation time for NPs. In addition, enzymatically active catalase in RBC membrane [169] could metabolize tumor endogenous H2O2 and ameliorate tumor hypoxia. PLGA-NPs coated with RBC membranes were engineered to co-deliver the chemotherapeutic agent curcumin, and the hypoxia-activated molecule, tirapazamine [170]. These drug-loaded and coated NPs induced apoptosis via generation of reactive oxygen species and consequent DNA damage suggesting the potential of the present system in circumventing hypoxic solid tumors. In another approach, the membrane from red blood cells was used to encapsulate nanoparticles consisting in perfluorocarbon inside PLGA [171]. This nanometic approach could provide efficient supply of oxygen. Recently, encapsulated Ag2S quantum dots in RBC membranes have been used as sonosensitizer to generate ROS under ultrasonic stimulation [172]. This design takes advantage of ultrasound to promote tumor blood flow improving hypoxic condition and enhancing sonotherapic effect of the system. In combination with oral anti-tumor drugs, this approach significantly increased survival of tumor-bearing mice.

Point 2. Paragraph 7 "Use of Nanoparticles for in vivo imaging of MSCs after use as anti-tumor cellular vehicle" is very short, not well positioned in the text logic course and thus has to be broadened and better integrated with the logic structure of the review.

Response 2. We agree with the reviewer that this paragraph was not well positioned. We have extended the text that is now included on pages 11-12, lines 444-466: Once the cells are transplanted in vivo, it is important to monitor the long-term fate of MSCs, their migratory capacity, their biodistribution, as well as their tumor penetration capacity. Conjugation of different types of nanoparticles to MSCs have successfully demonstrated feasibility to track migration and intratumor localization of MSCs by non-invasive imaging techniques used in preclinical and clinical settings such as magnetic resonance imaging (MRI), computed tomography (CT), ultrasound, optical imaging, positron emission tomography (PET) and single-photon emission computed tomography (SPECT). Adipose-derived MSCs loaded with mesoporous silica-coated manganese oxide NPs were efficiently monitored by MRI imaging over long periods after transplantation [126]. Adipose-derived MSCs can be also monitored by non-invasive CT imaging in vivo after labelling by PEG-coated gold NPs [127]. These gold NPs were visible at the transplantation site for as long as four weeks with no loss in signal. The authors were able to quantify the number of visualized cells as a function of the CT value obtained. These results are very important to quantify migratory and homing abilities of MSCs into tumor sites. Migratory capacity and tumor tropism toward malignant glioblastoma of both, bone marrow and placenta-derived MSCs was demonstrated by in vivo MRI tracking after labeling with superparamagnetic iron oxide (PEG-SPIO)-NPs [121, 128]. Although several nanoparticles have been designed for diagnostic and in vivo imaging, the optimal type of formulation for cell tracking in vivo does not, as yet, exist [129]. Further studies are necessary to use nanoparticles for diagnostic and imaging purposes in the field of oncology. The main challenges to be overcome are biocompatibility and improvement in the synthesis process. This approach would allow in vivo tracking and biodistribution of MSCs as carriers of therapeutic agents and would provide information about the tumor-targeted accumulation, drug release and long-term drug efficacy. Such information could contribute to a model of personalized medicine and patient individualization.

In addition, we have extended the in vivo imaging of exosomes after use as anti-tumor cellular vehicle on pages 12-13, lines 505-511: Fate of exosomes can be monitored in vivo by labeling them with NPs that are dectected by non-invasive imaging techniques such as magnetic resonance imaging (MRI), computer tomography (CT) or magnetic particle imaging (MPI) [144, 145]. These techniques have advantages and disadvantages according to their sensitivity, specificity, penetration, radiation and spatial resolution. Further developments need to be investigated to obtain more efficient, biocompatible and quantifiable exosome labeling and imaging techniques with the aim to translate exosome therapy to the clinic [144, 146].

Point 3. Furthermore, more schemes, figures/data taken from the cited papers as well as three resuming tables concerning each sub topic (MSC, exosomes, and Cell membrane-coated nanoparticles) and comparing in each cited references sub-case in terms of their use and the role of each component have to be inserted.

Response 3. According to the referee´s comment, we have included the following Tables and Figures:

Table 1. Nanotechnology strategies against hypoxia

Table 2. Advantages and disadvantages of the different cell-based delivery systems

Figure 1. Types of nanoparticles commonly used for biomedical applications.

Figure 2. Mesenchymal stem cell (MSC)-based strategies for targeted-cancer therapy.

Point 4. Other comments:

page 5, line 178-179: the authors should also mention the role played by ultrasound in helping drug or nanoparticles extravasation to reach the tumor site. In particular, the work of prof. Constantin Coussios should be mentioned. Here some examples:

Ultrasound Med Biol. 2011 Nov;37(11):1838-52. doi: 10.1016/j.ultrasmedbio.2011.08.004. Epub 2011 Oct 2. "Cavitation-enhanced extravasation for drug delivery".   Ultrasound Med Biol. 2019 Apr;45(4):954-967. doi: 10.1016/j.ultrasmedbio.2018.10.033. Epub 2019 Jan 14. "Microbubbles, Nanodroplets and Gas-Stabilizing Solid Particles for Ultrasound-Mediated Extravasation of Unencapsulated Drugs: An Exposure Parameter Optimization Study".

Response 4. According to the referee´s comment, the manuscript text has been modified to explain the role played by ultrasound and include the work of prof. Constantin Coussios.

Page 5, lines 170-178: Nanomedicine has taken advantage of this unique phenomenon, but extreme hypoxic condition in the central region of a large tumor mass can limit the EPR effect, and be a barrier for the entrance of NPs. Several methods have been reported to enhance EPR, such as, hyperthermia to mediate vascular permeabilization in solid tumors [53, 54], ultrasound-induced cavitation to modify tumor tissue [55, 56], application of nitric oxid (NO)-releasing agents to expand blood vessels [57] or administration of anti-hypertensives to normalize blood flow [58]. Some of these methods have been implemented from the field of nanomedicine to minimize side effects. Thus, different types of responsive-nanoparticles were designed to produce tumor heating after photostimulation, magnetism, radiofrequency waves or ultrasound [59].

Point 5. lines 279-285: it is fundamental to elucidate in the review the mechanisms of why MSC are not susceptible to the therapeutic content carried by the internalized NP while the cancer target cell are and if there are proofs and drawbacks of this method.

Response 5. According to the referee´s comment, we have discussed the effects of some common chemotherapy drugs on MSCs, as well as, how to overcome some of the drawbacks.

Pages 9-10, lines 346-367: MSCs can also incorporate small molecules of anti-tumor agents, such as paclitaxel or doxorrubicin, and carry them to tumor sites (Figure 2). However, this strategy has some downsides such as the low loading capacity and the rapid diffusional clearance of the molecules out of cells. Additionally, anti-cancer drugs may have some cytotoxic effects on MSCs and produce their loss before arrival at tumor sites. The effects of chemotherapeutics on MSCs have been quite controversial, from a reduction in proliferation and apoptosis, to resistance while retaining proliferation and differentiation potential (reviewed in [104]). MSCs are resistant to the cytotoxic effects of paclitaxel via inhibition of their proliferation, inhibition of apoptosis and induction of quiescence [105]. However, paclitaxel exposure does not up-regulate the expression of the trans-membrane pump P-glycoprotein 1 in MSCs, a mechanism by which cancer cells resist paclitaxel treatment [106]. Doxorrubicin at clinically used doses induces premature senescence of MSCs [107]. These senescent MSCs are functional but not proliferative, and are protected from doxorrubicin-induced tumor transformation. The effect of doxorubicin on MSCs in vivo is contradictory, from resistance to reduced proliferation rates and apoptosis [104], and this may be due to the ex vivo culture conditions of MSCs and duration of treatments. Hence, long-term in vitro and in vivo studies are necessary to understand the mechanisms behind the influence of chemotherapy on MSCs.

The encapsulation of chemotherapy drugs into NPs increases the drug-loading capacity of MSCs while reducing potential toxic effects on MSCs (Figure 2). In case of toxicity, the incorporation of controlled release or stimuli-responsive nanoparticles may avoid the loss of MSCs during the process of migration and tumor homing, as well as, ensuring that a therapeutic dose of the anti-cancer agent is released at the tumor site [108, 109].

Point 6. line 341: The authour mention just one example of encapsulation of Gold nanoparticles into exosomes for therapeutic and diagnostic purposes against cancer. Actually recently new papers were published about this topic using Metal Organic Framework (MOF) or Zinco oxide nanoparticels, which can be cited:

Chem. Mater. 2017, 29, 19, 8042-8046 "Exosome-Coated Metal–Organic Framework Nanoparticles: An Efficient Drug Delivery Platform"

Nanomedicine 2019 14:21, 2815-2833 "ZnO nanocrystals shuttled by extracellular vesicles as effective Trojan nano-horses against cancer cells"

Response 6. According to the referee´s comment, the manuscript text has been modified to include the new papers using Metal Organic Framework (MOF) or Zinc oxide nanoparticles, lines 493-495 (page 12): Efficient encapsulation of different types of nanoparticles into exosomes for therapeutic and diagnostic purposes in cancer is also possible [138-140].

Minor comments on English:

Point 7. line 181: change "to conduce" in "to conduct "

Response 7. This sentence was rewritten and that word has disappeared.

Point 8. line 379: change "such us2 in "such as"

Response 8. We apologize for these mistakes. We have corrected them.

Reviewer 2 Report

Really nice and sell written overview. Just revise the english grammar. Nixe work.

Author Response

Response to Reviewer 2 Comments

Comments and Suggestions for Authors

Really nice and sell written overview. Just revise the english grammar. Nixe work.

We really appreciate the positive comments made by the reviewer.

We apologize for any mistake in the English grammar in the previous version. English grammar has been reviewed again by a native speaker.

Reviewer 3 Report

Paz de la Torre et al present a very nice manuscript where they provide a systemic overview on cell-based nanoparticle delivery systems. Overall, this topic is very interesting and timely. Indeed, this is an interesting study. The manuscript is written in a clear way. However, I would propose to add discussed below parts to make the manuscript more elaborate and would improve the readability which certainly will attract broad readership.

At present the Introduction and subsequent major parts of the manuscript are not as scholarly and sceptical in their presentations of previous work as it should be. It is very appreciated that authors comment on limits of anti-angiogenic therapy. However, authors do not comment at all on side effects of nanoparticles and persistent problem of “off targeting”. I understand that this is not a primary goal of the manuscript, but in a review article one should include broad spectrum of the state of the art in the relevant field. A more thorough and careful presentation of what is and is not likely to be reproducible (i.e. true) in the field is essential.

Specifically, authors omitted described below studies and did not presented discussion of following papers. Those following publication should be included and discussion of the raised issues should be elaborated.

Targeting nanoparticles to malignant tissues for improved diagnosis and therapy is a popular concept. However, recent studied heavily criticized those approaches showing that only 0.7% (median) of the administered nanoparticle dose is found to be delivered to a solid tumour [1-4]. Those things need to be discussed and some outlook about liver delivery strategy should be proposed. This is very important, as far as this has negative consequences on the translation of nanotechnology for human use with respect to manufacturing, cost, toxicity, and imaging and therapeutic efficacy. Honest discussion is a way how to overcome future problems with overstatements. Authors very nicely discuss EPR effect and its integration in targeting. Somehow EPR effect may solve the problem of targeting particularly to liver cancer. However, although authors do discuss EPR targeting strategy, and importantly challenges associated with it are completely omitted [5-8]. Generally, there is a need for additional section/short paragraph for toxic/side effects of nanoparticles, like example [9-12]. Especially when one discusses polyethylenimine and silica nanoparticles. It would be nice to add a summary table of biological carriers that are used to deliver NPs highlighting specific crucial parameters of nanoparticles.

It is important to show together with prominent positive results also challenges and limitations. Because only in such a case it is possible to highlight them and maybe in future overcome. Hiding unwanted results will not help science development.

References

Wilhelm, S.; Tavares, A.J.; Dai, Q.; Ohta, S.; Audet, J.; Dvorak, H.F.; Chan, W.C.W. Analysis of nanoparticle delivery to tumours. Nature Reviews Materials 2016, 1, 16014. Shi, J.J.; Kantoff, P.W.; Wooster, R.; Farokhzad, O.C. Cancer nanomedicine: progress, challenges and opportunities. Nature Reviews Cancer 2017, 17, 20-37. Torrice, M. Does nanomedicine have a delivery problem? Acs Central Science 2016, 2, 434-437. Park, K. Facing the truth about nanotechnology in drug delivery. Acs Nano 2013, 7, 7442-7447. Hare, J.I.; Lammers, T.; Ashford, M.B.; Puri, S.; Storm, G.; Barry, S.T. Challenges and strategies in anti-cancer nanomedicine development: An industry perspective. Advanced Drug Delivery Reviews 2017, 108, 25-38. Navya, P.N.; Kaphle, A.; Srinivas, S.P.; Bhargava, S.K.; Rotello, V.M.; Daima, H.K. Current trends and challenges in cancer management and therapy using designer nanomaterials. Nano Convergence 2019, 6, 23. Hua, S.; de Matos, M.B.C.; Metselaar, J.M.; Storm, G. Current trends and challenges in the clinical translation of nanoparticulate nanomedicines: Pathways for translational development and commercialization. Frontiers in Pharmacology 2018, 9, 790. Rosenblum, D.; Joshi, N.; Tao, W.; Karp, J.M.; Peer, D. Progress and challenges towards targeted delivery of cancer therapeutics. Nature Communications 2018, 9, 1410. Lunova, M.; Prokhorov, A.; Jirsa, M.; Hof, M.; Olzynska, A.; Jurkiewicz, P.; Kubinova, S.; Lunov, O.; Dejneka, A. Nanoparticle core stability and surface functionalization drive the mTOR signaling pathway in hepatocellular cell lines. Scientific Reports 2017, 7, 16049. Lunova, M.; Smolkova, B.; Lynnyk, A.; Uzhytchak, M.; Jirsa, M.; Kubinova, S.; Dejneka, A.; Lunov, O. Targeting the mTOR signaling pathway utilizing nanoparticles: A critical overview. Cancers 2019, 11, 82. Patil, R.M.; Thorat, N.D.; Shete, P.B.; Bedge, P.A.; Gavde, S.; Joshi, M.G.; Tofail, S.A.M.; Bohara, R.A. Comprehensive cytotoxicity studies of superparamagnetic iron oxide nanoparticles. Biochem Biophys Rep 2018, 13, 63-72. Lewinski, N.; Colvin, V.; Drezek, R. Cytotoxicity of nanoparticles. Small 2008, 4, 26-49.

Author Response

Response to Reviewer 3 comments

Point 1. At present the Introduction and subsequent major parts of the manuscript are not as scholarly and skeptical in their presentations of previous work as it should be. It is very appreciated that authors comment on limits of anti-angiogenic therapy.

However, authors do not comment at all on side effects of nanoparticles and persistent problem of “off targeting”. I understand that this is not a primary goal of the manuscript, but in a review article one should include broad spectrum of the state of the art in the relevant field. A more thorough and careful presentation of what is and is not likely to be reproducible (i.e. true) in the field is essential.

Specifically, authors omitted described below studies and did not presented discussion of following papers. Those following publication should be included and discussion of the raised issues should be elaborated.

Targeting nanoparticles to malignant tissues for improved diagnosis and therapy is a popular concept. However, recent studied heavily criticized those approaches showing that only 0.7% (median) of the administered nanoparticle dose is found to be delivered to a solid tumour [1-4]. Those things need to be discussed and some outlook about liver delivery strategy should be proposed. This is very important, as far as this has negative consequences on the translation of nanotechnology for human use with respect to manufacturing, cost, toxicity, and imaging and therapeutic efficacy. Honest discussion is a way how to overcome future problems with overstatements. Authors very nicely discuss EPR effect and its integration in targeting. Somehow EPR effect may solve the problem of targeting particularly to liver cancer. However, although authors do discuss EPR targeting strategy, and importantly challenges associated with it are completely omitted [5-8]. Generally, there is a need for additional section/short paragraph for toxic/side effects of nanoparticles, like example [9-12]. Especially when one discusses polyethylenimine and silica nanoparticles.

References

1- Wilhelm, S.; Tavares, A.J.; Dai, Q.; Ohta, S.; Audet, J.; Dvorak, H.F.; Chan, W.C.W. Analysis of nanoparticle delivery to tumours. Nature Reviews Materials 2016, 1, 16014.

2- Shi, J.J.; Kantoff, P.W.; Wooster, R.; Farokhzad, O.C. Cancer nanomedicine: progress, challenges and opportunities. Nature Reviews Cancer 2017, 17, 20-37. 3- Torrice, M. Does nanomedicine have a delivery problem? Acs Central Science 2016, 2, 434-437.

4- Park, K. Facing the truth about nanotechnology in drug delivery. Acs Nano 2013, 7, 7442-7447.

5- Hare, J.I.; Lammers, T.; Ashford, M.B.; Puri, S.; Storm, G.; Barry, S.T. Challenges and strategies in anti-cancer nanomedicine development: An industry perspective. Advanced Drug Delivery Reviews 2017, 108, 25-38.

6- Navya, P.N.; Kaphle, A.; Srinivas, S.P.; Bhargava, S.K.; Rotello, V.M.; Daima, H.K. Current trends and challenges in cancer management and therapy using designer nanomaterials. Nano Convergence 2019, 6, 23.

7- Hua, S.; de Matos, M.B.C.; Metselaar, J.M.; Storm, G. Current trends and challenges in the clinical translation of nanoparticulate nanomedicines: Pathways for translational development and commercialization. Frontiers in Pharmacology 2018, 9, 790.

8- Rosenblum, D.; Joshi, N.; Tao, W.; Karp, J.M.; Peer, D. Progress and challenges towards targeted delivery of cancer therapeutics. Nature Communications 2018, 9, 1410.

9- Lunova, M.; Prokhorov, A.; Jirsa, M.; Hof, M.; Olzynska, A.; Jurkiewicz, P.; Kubinova, S.; Lunov, O.; Dejneka, A. Nanoparticle core stability and surface functionalization drive the mTOR signaling pathway in hepatocellular cell lines. Scientific Reports 2017, 7, 16049.

10- Lunova, M.; Smolkova, B.; Lynnyk, A.; Uzhytchak, M.; Jirsa, M.; Kubinova, S.; Dejneka, A.; Lunov, O. Targeting the mTOR signaling pathway utilizing nanoparticles: A critical overview. Cancers 2019, 11, 82.

11- Patil, R.M.; Thorat, N.D.; Shete, P.B.; Bedge, P.A.; Gavde, S.; Joshi, M.G.; Tofail, S.A.M.; Bohara, R.A. Comprehensive cytotoxicity studies of superparamagnetic iron oxide nanoparticles. Biochem Biophys Rep 2018, 13, 63-72.

12- Lewinski, N.; Colvin, V.; Drezek, R. Cytotoxicity of nanoparticles. Small 2008, 4, 26-49.

Response 1. We appreciatively accept the comments made by the reviewer, aimed to improve the quality of the review article. We agree with the reviewer that is important to discuss about challenges and limitations in addition to the positive results. We have added a new section entitle “The limits of nanomedicine in clinical applications”. We discuss the limits clinical translation of nanomedicine for the treatment of cancer, the safety and toxicity of NPs in humans, the “off target” location of nanoparticles in healthy tissues, the existence of the EPR phenomenon and why only a small percentage of systemically injected NPs accumulate in tumors (a median of 0.7%), and the problems of actively targeted NPs. This section is located in pages 6-7, lines 217-282:

The limits of nanomedicine in clinical applications

It is a fact that clinical translation of nanomedicine for the treatment of cancer remains a great challenge. Regardless of important contributions of nanotechnology to oncology in minimizing toxic side effects of drugs, overall survival of patients has not improved. Several relevant questions must be addressed in order to improve applicability of nanomedicine formulations to treat cancer, and this requires the understanding of the complexity and heterogeneity of human tumors and a deeper insight into nano-bio interactions.

For NPs to have clinical translation potential there is a need to evaluate their safety and toxicity in humans and determine how large-scale manufacturing processes can introduce changes in this profile. Although safety of many materials has been proven, as the complexity of nanoparticles increases by the use of synthetic compositions or by addition of ligands or coatings, the in vivo behavior and the toxicological profile must be evaluated. The main safety concerns can derive from direct cell toxicity, nanoparticles aggregation, long-term accumulation, hemolytic effects, and/or immunogenic behavior [69]. Toxicological evaluation of nanoparticles is based on the understanding of their in vivo distribution, metabolism and excretion [70].

To take advantage of nanomedicine, it is vital to optimize nanomaterial properties such as drug loading capacity and/or ability of sustained release of the cargo in vivo, among others. Furthermore, it is essential to minimize the location of nanoparticles in healthy tissues and improve the delivery to the target organ. Increasing the efficiency in the delivery of nanoparticles to the tumor is considered the main goal in order to achieve real benefit [71].

The use of nanomedicine in cancer therapy has been supported by the existence of the EPR phenomenon; however, only a small percentage of systemically injected NPs accumulate in tumors (a median of 0.7% according to a wide meta-analysis study based on preclinical data)[72]. EPR seems to be an overestimated effect as its understanding is based on the high EPR existing in fast-growing subcutaneous tumor xenografts in mice models. However, non-invasive imaging techniques applied to a small number of patients to determine the penetration and accumulation of nanoparticles in tumors, revealed that EPR is not a uniformly extended effect in solid tumors in humans [67]. Variability in vascular permeability, blood velocity, interstitial blood pressure, oncotic pressure, and complexity of tumor stroma influence the movement of nanoparticles into and out of the tumor [73]. Additionally, physicochemical properties of nanoparticles, principally size and shape, also affect NPs extravasation and accumulation. In order to predict tumor susceptibility to EPR, and therefore, to benefit from the use of nanomedicine, some attempts have been made to characterize EPR related genes, proteins or cell biomarkers (reviewed in [74]). Several studies have suggested the value of stratifying subpopulations of cancer patients according to their EPR relevance, in order to define “right patients” to be treated by nanomedicine strategies, in an analogous manner as is being done in the development of other anti-cancer strategies [67].

As an alternative to passive accumulation, active targeting of nanoparticles is proposed in order to improve their tumor retention and to favor their uptake by the target cells. This strategy relies on the interaction between ligands conjugated onto the surface of nanoparticles (e.g. antibodies, peptides or carbohydrates) and their target. Target substrates can be surface receptors expressed by tumor cells or by other cells in the tumor microenvironment, secreted molecules or even the physicochemical environment in the tumor. An additional advantage of actively targeted NPs could lie in their capacity to target disseminated locations throughout the body, such as metastatic lesions or hematological cancers where EPR does not exist [75]. However, several problems have made the use of ligand-targeted approaches at the clinical level, so far, negligible. These problems are the accessibility and expression of the target, the anatomical and physiological barriers to NPs delivery, as well as, the lack of real knowledge about toxicities of these complex formulations [70].

Targeting of nanoparticles to tumors, whether active or passive, must overcome physiological barriers to reach the tumor site once systemically administered. Whatever increases circulating lifetime by reducing clearance means an improvement in efficacy. Clearance of NPs by kidneys and their sequestration by reticuloendothelial organs are the main barriers affecting their bio-distribution, and therefore must be considered at the design stage. Renal elimination of nanoparticles is determined by their size, charge, shape and surface composition [76]. Recognition of NPs by immune cells and retention by the reticuloendothelial organs, such as liver, spleen or bone marrow, constitute the other major obstacle to the success of nanoparticle delivery since they lead to a premature elimination from the bloodstream. Having interacted with biological fluids, nanoparticles are exposed to active biomolecules and diverse serum proteins non-specifically adhere onto their surface forming a protein corona. There is an evident impact of protein corona in the fate and biological effects of nanoparticles [77, 78].

Several surface-coating molecules such as polyethylene glycol (PEG) have been used to provide “stealth” properties to NPs during circulation [79]. Nonetheless, complement-related responses to PEG result in mild to severe hypersensitivity reactions in some susceptible individuals [80]. In addition, PEG-specific antibodies have been detected after repeated administration of PEG-coated liposomes in the same animal [81]. These immunological responses may lead to altered pharmacokinetics and subsequent loss of efficacy of the treatment, and to potentially serious toxicities including anaphylaxis.

Point 2. It would be nice to add a summary table of biological carriers that are used to deliver NPs highlighting specific crucial parameters of nanoparticles.

Response 2. According to the referee´s comment, we have included the following Tables and Figures:

Table 1. Nanotechnology strategies against hypoxia

Table 2. Advantages and disadvantages of the different cell-based delivery systems

Figure 1. Types of nanoparticles commonly used for biomedical applications.

Figure 2. Mesenchymal stem cell (MSC)-based strategies for targeted-cancer therapy.

Point 3. It is important to show together with prominent positive results also challenges and limitations. Because only in such a case it is possible to highlight them and maybe in future overcome. Hiding unwanted results will not help science development.

Response 3. We agree with the reviewer that is important to discuss about challenges and limitations in addition to the positive results. Because of that, in addition to all the comments included in the previous version, we have incorporated some additional phrases to emphasize that cancer treatment by nanomedicine has many challenges to face despite the good results obtained in in vitro and preclinical studies. Here are some examples:

Page 5, lines 170-178: Nanomedicine has taken advantage of this unique phenomenon, but extreme hypoxic condition in the central region of a large tumor mass can limit the EPR effect, and be a barrier for the entrance of NPs. Several methods have been reported to enhance EPR, such as, hyperthermia to mediate vascular permeabilization in solid tumors [53, 54], ultrasound-induced cavitation to modify tumor tissue [55, 56], application of nitric oxid (NO)-releasing agents to expand blood vessels [57] or administration of anti-hypertensives to normalize blood flow [58]. Some of these methods have been implemented from the field of nanomedicine to minimize side effects. Thus, different types of responsive-nanoparticles were designed to produce tumor heating after photostimulation, magnetism, radiofrequency waves or ultrasound [59].

Pages 7-8, lines 288-296: Nanomedicine appears as a valuable tool to achieve these goals having the opportunity to design polyvalent NPs. Unfortunately, physiological barriers decrease NPs circulating lifetime and hinder their delivery to the tumor site. Additionally, the hypoxic region of the tumor constitutes an insuperable barrier resulting in an inefficient distribution of the NPs, and as a consequence, a non-uniform release of drugs into the tumor. Furthermore, potential toxicity of NPs, owing to their composition and/or to the nano-bio interactions, can compromise feasibility of their use [82]. It is necessary to find solutions to overcome these problems without forgetting the great heterogeneity of human tumors. Encapsulation of nanoparticles into cell- or cell membrane- based systems can enable the address of these issues to some extent.

Page 8, lines 302-305: As discussed below, mesenchymal stem cells and their exosomes have tumor tropism because tumor hypoxia is a potent mediator directing MSCs migration [83]. Although the use of these encapsulated formulas is in their initial stages, they appear as a potential way to reach the impenetrable hypoxic core of solid tumors.

Page 9, lines 343-345: The use of viral and non-viral vectors for genetic modifications of MSCs has several drawbacks, such as transient gene expression and low transfection efficiency, along with a high risk of tumor cell transformation [103].

Page 10, lines 371-373: Although, progress have been made in the design of NPs to introduce anti-cancer drugs to be transported by MSCs, this combined system MSC/NP is still in its initial stages.

Page 12-13, lines 508-511: These techniques have advantages and disadvantages according to their sensitivity, specificity, penetration, radiation and spatial resolution. Further developments need to be investigated to obtain more efficient, biocompatible and quantifiable exosome labeling and imaging techniques with the aim to translate exosome therapy to the clinic [144, 146].

Page 14, lines 577-578: Although encouraging results have been found using cell membrane-coated nanoparticles to treat tumor angiogenesis and hypoxia, this field needs further lines of investigation

Page 14, lines 607-610: Besides the challenges of biocompatibility and improvement in the synthesis process of NPs, further studies are necessary to unravel the role of MSCs facilitating or inhibiting tumor growth. Therefore, the use of the MSC / NP system needs additional studies before it can be used clinically in humans.

Round 2

Reviewer 1 Report

The authors have provided extensive review on their manuscript.

Reviewer 3 Report

Authors did all necessary changes to the text. They did very significant work to improve the readability of the manuscript. Their responses are very appreciated. The manuscript is ready for publication.